# P62/SQSTM1 binds with claudin-2 to target for selective autophagy in stressed intestinal epithelium

Rizwan Ahmad[1,5], Balawant Kumar[1,5], Raju Lama Tamang[1], Geoffrey A. Talmon[2], Punita Dhawan[1,3,4] & Amar B. Singh ● [1,3,4 ✉]

Impaired autophagy promotes Inflammatory Bowel Disease (IBD). Claudin-2 is upregulated in IBD however its role in the pathobiology remains uncertain due to its complex regulation, including by autophagy. Irrespective, claudin-2 expression protects mice from DSS colitis. This study was undertaken to examine if an interplay between autophagy and claudin-2 protects from colitis and associated epithelial injury. Crypt culture and intestinal epithelial cells (IECs) are subjected to stress, including starvation or DSS, the chemical that induces colitis in-vivo. Autophagy flux, cell survival, co-immunoprecipitation, proximity ligation assay, and gene mutational studies are performed. These studies reveal that under colitis/stress conditions, claudin-2 undergoes polyubiquitination and P62/SQSTM1-assisted degradation through autophagy. Inhibiting autophagy-mediated claudin-2 degradation promotes cell death and thus suggest that claudin-2 degradation promotes autophagy flux to promote cell survival. Overall, these data inform for the previously undescribed role for claudin-2 in facilitating IECs survival under stress conditions, which can be harnessed for therapeutic advantages.

[1] Department of Biochemistry and Molecular Biology, University of Nebraska Medical Center, Omaha, NE, USA. [2] Department of Pathology and Microbiology, University of Nebraska Medical Center, Omaha, NE, USA. [3] Fred and Pamela Buffett Cancer Center, University of Nebraska Medical Center, Omaha, NE, USA. [4] VA Nebraska-Western Iowa Health Care System, Omaha, NE, USA. [5] These authors contributed equally: Rizwan Ahmad, Balawant Kumar. ✉email: amar.singh@unmc.edu

An impaired intestinal epithelial barrier is a hallmark of inflammatory bowel disease (IBD)[1]. Besides environmental factors, genetic and molecular alterations lead to the gut barrier dysregulation and intestinal inflammation, including in IBD patients[2]. In addition to the stoichiometric alternations in the barrier constituent proteins, epithelial wound healing processes, dependent on proliferation, migration, differentiation, and survival of intestinal epithelial cells (IECs), counteract the breakdown in the gut barrier integrity[3–6]. Autophagy plays a key role in regulating IECs homeostasis and renders protection against stress-induced cell death, including inflammatory conditions[7]. Autophagy is impaired in IBD and has pleiotropic implications in gut homeostasis, including IECs survival[8–11]. However, molecular undertakings of how autophagy renders protection to IECs and thus preserve the gut barrier integrity are not well understood.

Tight junctions (TJ) play a key role in maintaining the gut barrier integrity, and accordingly proteins constituting the TJ structure/function are dysregulated in IBD[2]. In this regard, studies including ours have shown a robust upregulation in claudin-2 expression in IBD[12–14]. Claudin-2 is also known as the leaky claudin due to its role in the paracellular cation transport[15,16]. However, the causal role of claudin-2 in IBD remains uncertain as genetic manipulation of claudin-2 has context-specific and contrasting effects in murine models of IBD[2,17–19]. Nevertheless, genetic modeling of pathological claudin-2 expression, as in IBD, protects mice from colitis induced by epithelial injury while claudin-2 KO mice develop severe colitis[17–19]. Claudin-2 expression in the IECs is inhibited by starvation-induced autophagy, which is postulated to promote barrier integrity due to the role of claudin-2 as a leaky claudin[20]. However, autophagy plays critical role in promoting cell survival, particularly under stress conditions. Therefore, based on the findings that claudin-2 KO mice develop severe colitis and epithelial injury when subjected to colitis, we postulated that autophagy targets claudin-2 to protect the IECs from stress-induced death and thus preserve the barrier integrity.

To test our hypothesis, we subjected IECs and 3D-culture of mouse colon crypts from WT and claudin-2 manipulated mice to commonly used colitogen (DSS) or starvation. These stimuli induced autophagy and downregulated claudin-2 expression. Inhibiting autophagy prevented both, colitogen or starvation-induced decrease in claudin-2 expression and promoted cell death. Inhibiting autophagy under conditions of claudin-2 KO exacerbated the cell death compared to respective controls. Further analysis using immunoprecipitation, antigen-depletion and proximity-ligation studies showed that claudin-2 is poly-ubiquitinated and physically binds with P62/SQSTM1. Most notably, mutating lysine 218 in claudin-2 cytoplasmic tail prevented the colitogen or starvation-induced decrease in claudin-2 and promoted cell death. Silencing ATG16L expression similarly prevented the colitogen/stress-induced decrease in claudin-2 and promoted cell death. Taken together, our data identifies claudin-2 as a selective substrate for P62/SQSTM1-assisted autophagy and its role in promoting survival against colitis-induced epithelial cell death to improve the gut barrier integrity.

## Results

### Colitogens, known to induce intestinal epithelial injury and/or colitis, downregulate intestinal epithelial claudin-2 expression.

In face of the published data from our laboratory, and of others, that Villin-claudin-2 transgenic (TG) mice are protected from DSS-colitis while claudin-2 KO mice develop severe colitis and epithelial injury[17–19], we wondered if claudin-2 expression renders protection to IECs against colitis-induced cell death. Confluent culture of IECs (Caco-2 and HT29 cells) were subjected to DSS treatment (2.5% w/v in complete culture medium). Interestingly, DSS

treatment resulted in sharp decreases in claudin-2 expression in both cell lines in dose- and time-dependent manners (Fig. 1a, b). In same lysates, expression of E-cadherin or claudin-4 did not change and thus supported specificity of the changes in claudin-2 expression. We found a similar decrease in claudin-2 expression using lysates prepared from the distal colon from mice subjected to DSS-colitis (2.5% w/v) for 7 days (Fig. 1c). Additional analysis validated that the DSS treatment induced growth arrest, DNA damage and apoptosis (Supplementary Fig. 1a–h). We found a similar downregulation of claudin-2 expression in Caco-2 cells exposed to 2,4,6-Trinitrobenzenesulfonic acid (TNBS), also used to induce colitis in vivo (Fig. 1d). Starvation, yet another type of stress, also induced downregulation of claudin-2 expression (Fig. 1e). In sum, above results suggested that intestinal claudin-2 expression is downregulated under conditions of IECs stress/injury.

### Colitogen-induced claudin-2 downregulation is largely post-transcriptional and preventable by inhibiting autophagy.

To understand whether claudin-2 downregulation in above studies is due to the regulation at mRNA or protein expression, we performed qRT-PCR and immunoblot analysis. Interestingly, claudin-2 protein as well as mRNA expressions were downregulated however the decrease in claudin-2 protein expression was significantly higher (Supplementary Fig. 2a). To further clarify, we subjected Caco-2 cells to DSS treatment alone or with Actinomycin-D (Act-D), an inhibitor of mRNA synthesis, or Cycloheximide (CHX), a protein synthesis inhibitor. As shown in Fig. 2a, co-treatment of the DSS-treated cells with CHX but not Act-D showed the maximum decrease in claudin-2 expression. Taken together, these data suggested post-translational regulation as major regulatory mechanism of intestinal claudin-2 regulation under stress/injury. In light of this data and the fact that claudin-2 protein is degraded in the lysosomes[20], we examined whether Chloroquine (CHLQ), an inhibitor of lysosomal degradation of proteins would restore the DSS-dependent decreases in claudin-2 expression. As shown in Fig. 2b, co-treatment of IECs with CHLQ and DSS largely rescued the DSS-dependent decrease in claudin-2 levels. Similar results were obtained upon the use of Bafilomycin A1 (Baf A1), yet another inhibitor of lysosomal protein degradation (Fig. 2b)[20]. Similar to the colitogen-dependent decrease, the starvation-induced decrease in claudin-2 expression was also restored upon co-treatment with CHLQ or Baf A1 (Fig. 2c). Co-immunofluorescence analysis using anti-claudin-2 and anti-Lamp-1 antibodies further verified lysosomal degradation of claudin-2 in DSS-treated cells (Fig. 2d). Overall, these results suggested post-translational regulation of claudin-2 and its lysosomal degradation as a major regulatory mechanism under the conditions of intestinal epithelial stress/injury.

### DSS treatment induces autophagy.

To explain the above findings, we examined whether autophagy flux is also upregulated in colitogen-treated cells and mice. We used acridine-orange staining and overexpression of a GFP-LC-3-RFP-LC-3ΔG autophagy reporter construct, commonly used to detect the dynamism of autophagy flux[21,22]. A significant increase in the formation of orange puncta (acridine-orange staining) or more yellow (auto-lysosome) and red (autophagosome) LC-3 puncta in DSS-treated cells compared to control cells suggested an increase in autophagic flux (Fig. 2e, f). To validate these findings, we further determined whether there is a progressive increase in the lipidation of LC-3 I to LC-3 II. Both, Caco-2 cells and mouse colon lysate were used. We found a dose and time-dependent increase in the lipidation of LC-3 II in DSS-treated cells, which was inhibited upon Baf A1 treatment (Fig. 2g–i). Similar increase in LC-3 II expression was found in the colons of mice subjected to DSS-colitis (Fig. 2j). However, claudin-2 protein is trafficked through the endosomal-lysosomal

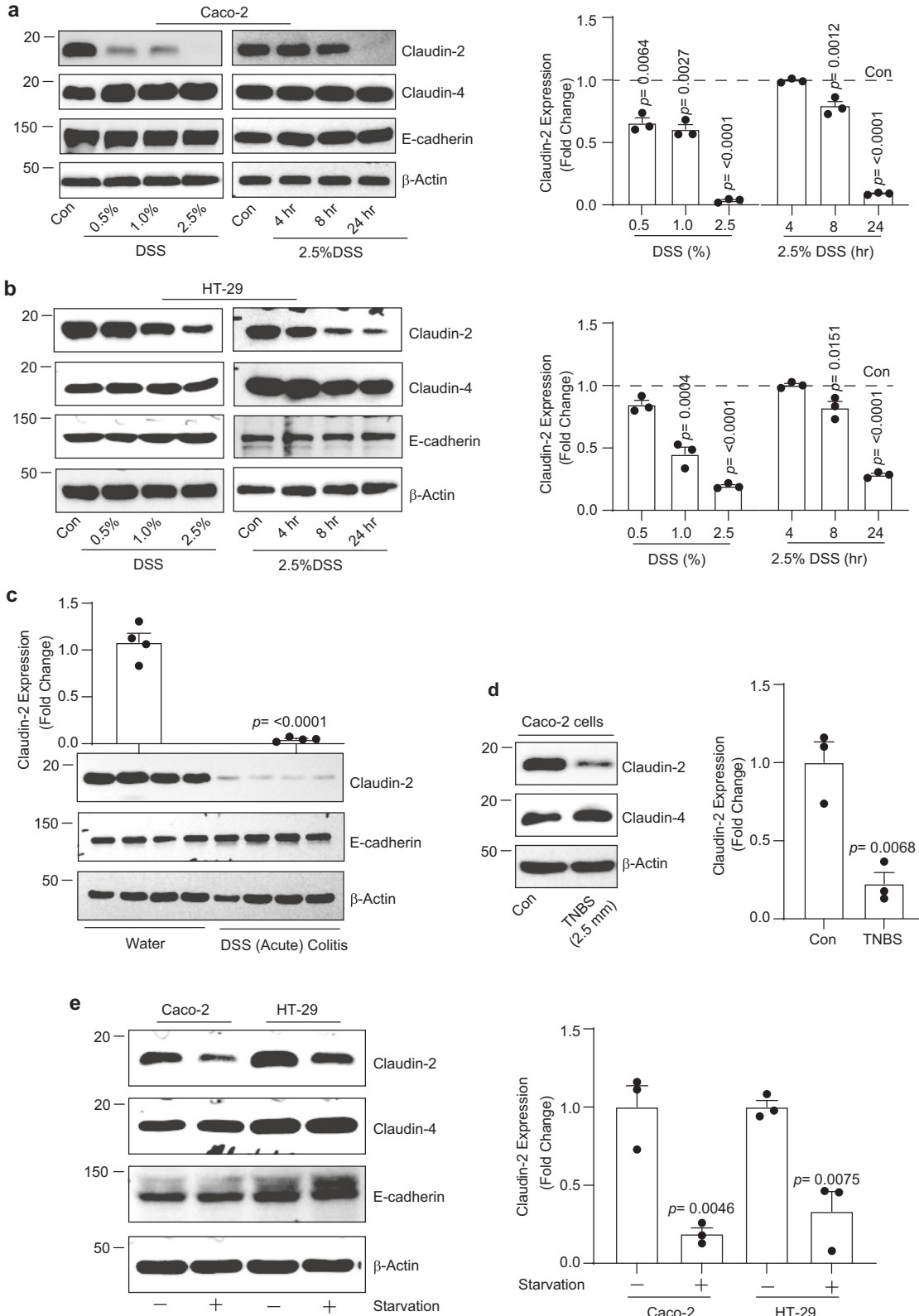

pathway[23,24]. Hence, to exclude the potential that the interference with normal claudin-2 trafficking has contributed towards above outcome, we further determined effects of inhibiting endosomal and proteasomal protein degradation along with lysosomal degradation. To account for the baseline trafficking/degradation, we subtracted the changes in claudin-2 expression in control cells subjected to respective inhibitors with the changes in the

DSS-treated cells. As demonstrated in Supplementary Fig. 2b, c, accumulation of claudin-2 protein in cells co-treated with DSS and inhibitors of lysosomal/autophagy degradation was significantly higher compared to the control cells as well as cells subjected to the co-treatment with DSS + YM201636 (an inhibitor of endosomal protein degradation) or MG132 (an inhibitor of proteasomal protein degradation). Taken together, our data from

**Fig. 1 In intestinal epithelial cells, colitogen, or nutrient deprivation (starvation) downregulates claudin-2 expression.** Exposure to DSS inhibited claudin-2 expression in IECs in time- and dose-dependent manners. **a** Immunoblotting and densitometric analysis using Caco-2 cells and **b** HT29 cells ($n = 3$ independent experiments. Results are presented as means ± SEM. P values from one-way ANOVA followed by Tukey's post hoc test). **c** Immunoblotting and densitometric analysis for claudin-2 expression using the lysates from the distal colon from mice subjected to DSS (Acute) colitis (2.5% DSS w/v in drinking water for 7 days; $n = 4$ mice/group and results are presented as means ± SEM. P values from Student's t test). E-cadherin expression served as an epithelial cell marker. **d** Immunoblotting and densitometric analysis for claudin-2 and claudin-4 expression in Caco-2 cells exposed to TNBS (2.5 mM) for 24-h ($n = 3$ independent experiments. Results are presented as means ± SEM. P values from Student's t test.) in complete culture medium. **e** Immunoblotting and densitometric analysis for claudin-2 and claudin-4 expression in IECs subjected to nutrient deprivation/starvation ($n = 3$ independent experiments. Results are presented as means ± SEM. P values from Student's t test).

both, colitogen-treated IECs and mice suggested that colitogen induce an increase in the autophagy flux and degradation of claudin-2 protein, which was preventable using inhibitors of lysosomal degradation/autophagy.

**Claudin-2 physically associates with P62/SQSTM1.** The above findings suggested that stress downregulates intestinal claudin-2 expression by promoting its degradation *via* autophagy. In determining how autophagy mediates claudin-2 degradation, we focused on P62/SQSTM1 as it plays a critical role in autophagic degradation of cellular proteins by serving as a scaffolding protein[25]. Co-immunofluorescence analysis revealed that claudin-2 co-localizes with both, P62/SQSTM1 and LC-3 (Fig. 3a, b). To determine if claudin-2 physically interacts with P62/SQSTM1, we performed immunodepletion (Co-immunoprecipitation; Co-IP) using anti-claudin-2 antibody followed by immunoblotting with anti-P62/SQSTM1 antibody, and vice versa. As shown in Fig. 3c, d, immunodepletion with anti-claudin-2 or P62/SQSTM1 antibody was able to pull down the other protein, which increased with increasing amounts of the respective antibody used for the immunodepletion. To validate, we performed a complimentary dose-dependent antigen-antibody capture or depletion assay using claudin-2 or P62/SQSTM1 antibody, as described in the Materials and Methods. The antibody-mediated depletion of claudin-2 from the cell lysates also resulted in a progressive loss of P62/SQSTM1 from the cell lysates (Fig. 3c). Using anti-P62/SQSTM1 antibody for immunodepletion, we found a similar pulldown for claudin-2 protein or its depletion from the cell lysate (Fig. 3d). Based on these findings, we postulated that claudin-2 binding with P62/SQSTM1 for sequestration and autophagic degradation should be an early event in the stressed IECs. Therefore, we exposed Caco-2 cells to 2.5% DSS for 2 h and repeated the Co-IP analysis. As shown in Fig. 3e, the association between claudin-2 and P62/SQSTM1 was increased in DSS-treated cells compared to the control cells. Proximity-ligation assay (PLA) done under similar experimental conditions further demonstrated colocalization of claudin-2 and P62/SQSTM1 (Fig. 3f). To validate a causal role for P62/SQSTM1 in stress/colitogens-mediated claudin-2 degradation, we genetically inhibited (siRNA-mediated silencing) expression of P62/SQSTM1. As shown in Fig. 3g, the loss of P62/SQSTM1 expression prevented both, the DSS- and starvation-mediated decrease in claudin-2 expression (Fig. 3g). Overall, these data suggested that claudin-2 physically binds with P62/SQSTM1, which in turn targets claudin-2 protein for autophagy-mediated degradation and this association increases under conditions of stress.

**Colitogen/stress induces lysine (K)63-linked ubiquitination of claudin-2 for its autophagic degradation.** An increasing body of evidence suggests that the polyubiquitinated protein cargoes are marked for degradation[26]. P62/SQSTM1 interacts noncovalently with ubiquitin or polyubiquitin chains to deliver the polyubiquitinated cargoes to the autophagosomes[27,28]. We therefore wondered if claudin-2 is ubiquitinated in the stressed IECs, which is then sequestered by P62/SQSTM1 for selective autophagy.

To test, we used PYR41, a cell-permeable ubiquitin-activating enzyme inhibitor[29]. Caco-2 cells were subjected to DSS (2.5% w/v) treatment or nutrient starvation with or without PYR41. Immunoblot analysis using the lysates from these cells demonstrated that inhibiting ubiquitination prevented the colitogen or starvation-induced claudin-2 downregulation (Fig. 4a, b). To further inquire into these findings, we performed Co-IP analysis using a pan-ubiquitin and claudin-2 antibody. As shown in Fig. 4c, immunoblot analysis using the immunoprecipitants from the Co-IP using a pan-ubiquitin antibody pulled down claudin-2 and thus demonstrated that claudin-2 protein is ubiquitinated. Immunofluorescent analysis in complimentary proximity-ligation (PLA) analysis performed using claudin-2 and pan-ubiquitin antibodies further demonstrated colocalization (Fig. 4d).

Notably, ubiquitination is involved in both, the ubiquitin-proteosome system of protein degradation and autophagy[30]. Based on the lysine residues inside the ubiquitin, seven homogeneous polymer chain linkages can be defined: K6, K11, K27, K29, K33, K48, and K63 where K63 is a common ubiquitination marker in the autophagic protein degradation process[31–33]. Subsequently, we examined whether claudin-2 protein undergoes K63-linked ubiquitination. Co-IP analysis was done using antibodies against claudin-2 and K63-ubiquitin using DSS-treated cell lysates. We also examined possible association of claudin-2 with the K48 ubiquitin. The outcome from the Co-IP is presented in Supplementary Fig. 3 and Fig. 4e which showed that the DSS treatment induced both, K63 and K48-linked polyubiquitination. We also found minor interaction of claudin-2 with K48 ubiquitin; however, the interaction of claudin-2 with K63-ubiquitin was the predominant interaction. Complimentary PLA analysis further showed colocalization of claudin-2 with K63-linked ubiquitin (Fig. 4f). Taken together, our data suggested that colitogens or stress-induced K63-linked ubiquitination of claudin-2 protein.

**K218 in claudin-2 cytoplasmic tail is important for its K63-linked ubiquitination and Colitogen or stress-induced degradation.** We next examined specific lysine residue in claudin-2 protein which is responsible for the colitogens or stress-induced ubiquitination and degradation. Using an online software (http://bdmpub.biocuckoo.org/prediction.php), we identified two highly probable lysine sites for ubiquitination in claudin-2 cytoplasmic tail (K216 and K218) (Fig. 5a and Supplementary Table 1). To examine their functional relevance, we substituted these lysine residues with alanine. A C-terminal HA-tag was added to the resultant claudin-2 mutant constructs to differentiate the mutant claudin-2 proteins from endogenous claudin-2 protein (Supplementary Table 2). Thereafter, full-length (Cldn2-HA), K216A, K218A and double mutant (K216AK218A) claudin-2 expression plasmid constructs were overexpressed in Caco-2 cells and effect of DSS treatment or starvation was examined. The full-length claudin-2 protein and the K216A mutant claudin-2 protein were significantly downregulated in response to both, the DSS treatment or starvation. The K218A mutant claudin-2 protein did not undergo similar downregulation in response to exposure to either DSS or starvation. Claudin-2

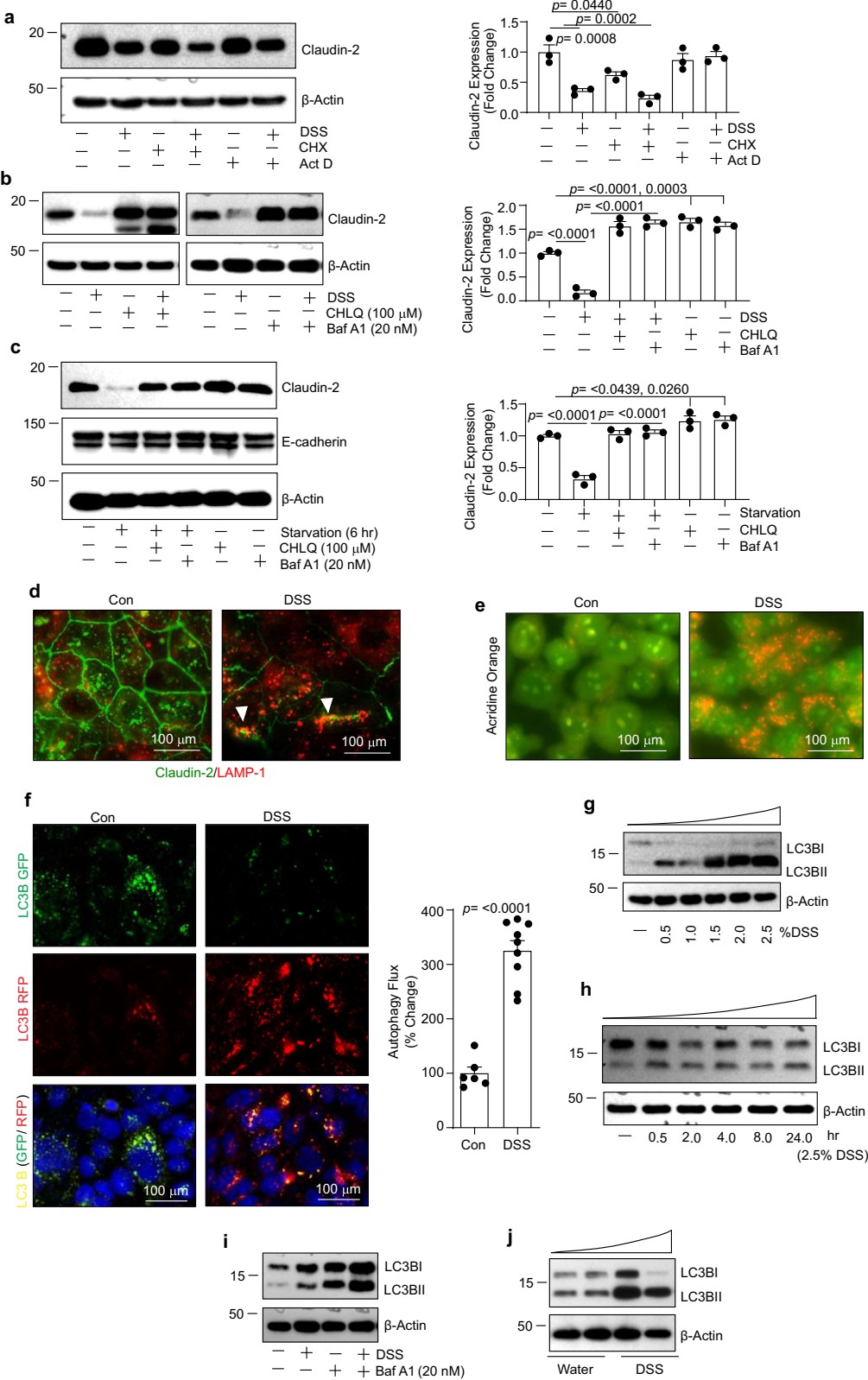

mutant protein where lysine at both, 216 and 218 residues were substituted showed an intermediate level of expression (Fig. 5b and Supplementary Fig. 4a–c).

The above results also implicated that the mutant K218A claudin-2 mutant protein would not bind with the K63-linked ubiquitin, which we further tested using Co-IP studies. As shown in Fig. 5c, the immunoblotting using the immunoprecipitants

from the Co-IPs using anti-K63-linked ubiquitin and anti-HA-tag antibodies showed the association of K63 ubiquitin with both, the full-length and K216A mutant claudin-2 proteins but not with K218A mutant claudin-2 protein. The similar outcome we got when we Co-IP studies using the lysates of DSS-treated cells further showed that the K218A mutant claudin-2 did not bind with the K63-ubiquitin with DSS while full-length or K216A

**Fig. 2 Colitogen/stress promote autophagy and claudin-2 protein degradation. a** Immunoblotting and densitometric analysis using Caco-2 cell lysate subjected to DSS treatment with or without co-treatment with Actinomycin-D (Act-D) or Cycloheximide (CHX) ($n = 3$ independent experiments. Results are presented as means ± SEM. *P* values from one-way ANOVA followed by Tukey's post hoc test). **b** Immunoblot analysis using total cell lysates. Caco-2 cells were subjected to DSS treatment with or without Chloroquine (100 μM) or Bafilomycin A1 (20 nM). $n = 3$ independent experiments. Results are presented as means ± SEM. *P* values from one-way ANOVA followed by Tukey's post hoc test. **c** Immunoblotting and densitometry analysis using total cell lysates of Caco-2 cells subjected to starvation with/without Chloroquine or Bafilomycin A1 ($n = 3$ independent experiments. Results are presented as means ± SEM. *P* values from one-way ANOVA followed by Tukey's post hoc test). **d** Representative images of immunofluorescent co-staining for claudin-2 and LAMP1 in control and DSS-treated Caco-2 cells (for 24 h). Arrows indicate colocalization of claudin-2 and LAMP1. **e** Representative images of Acridine-orange staining in Caco-2 cell treated with DSS (2.5%) (at 24 h post-treatment). **f** Representative images and autophagy flux rate of LC-3-GFP/RFP reporter construct in control and DSS-treated Caco-2 cells. $n = 3$ independent experiments. Results are presented as means ± SEM. *P* values from Student's *t* test. **g–i** Immunoblot analysis for relative expression of LC-3 I and LC-3 II. Lysates from DSS-treated Caco-2 cells (dose- and time-dependent) as well as using Baf A1 pre-treatment were used, respectively. **j** Immunoblot analysis for relative expression/lipidation of LC-3 I to LC-3 II in colon tissue lysate from mice subjected to acute (DSS) colitis.

mutant did. Here, we used the DSS and DSS + PYR41-treated cell lysate as control for K63-linked ubiquitination. In accordance with the above results, the K218A mutant claudin-2 protein also did not degrade in response to the DSS treatment (Fig. 5d). Collectively, above findings validated that K63-linked ubiquitination at K218 facilitated selective autophagic targeting of claudin-2 under stress conditions including colitis.

**Inhibiting autophagy-mediated claudin-2 degradation promotes cell death in stressed IECs.** The ubiquitin-mediated selective autophagy promotes cell survival[34,35]. Claudin-2 over-expression protects mice from DSS-colitis and associated epithelial injury[17,18]. We therefore postulated that in the stressed intestinal epithelium, claudin-2 degradation by selective autophagy helps promote autophagy flux and thus cell survival. To test, we subjected Caco-2 cells to DSS treatment with or without autophagy inhibitors (CHLQ or Baf A1). Effects on cell survival and claudin-2 expression were determined. As expected, inhibiting autophagy prevented the DSS-induced claudin-2 degradation (Fig. 2b) while promoted cell death (Fig. 6a). To examine, if inhibiting autophagy in vivo would have similar effects, we inhibited autophagy in mice. Normal mice (C57BL/6; 6–8 weeks old) were orally administered (gavage) with 36-077 (40 mg/kg body weight/day), a highly selective and potent inhibitor of autophagy (an inhibitor of VPS34/PIK3C3)[36]. Mice were sacrificed after 3 days, and total tissue lysates were prepared. Immunoblot analysis was done using the total colon lysates. Cleaved caspase-3 expression was determined as a marker of cell death. As expected, claudin-2 protein was markedly upregulated/ accumulated in the colons of the mice subjected to 36-077-treatment. The expression of cleaved caspase-3 was also upregulated in these mice colons (Fig. 6b). These data supported the assumption that claudin-2 degradation under stress promotes IECs survival potentially by promoting autophagy flux.

In light of the above data, we postulated that inhibiting autophagy as in IBD patients should prevent colitogen-induced claudin-2 protein degradation and promote cell death. ATG16L, a key driver of autophagy is ubiquitous in its expression and often mutated in IBD patients[8]. Thus, we performed siRNA-based silencing of ATG16L in Caco-2 cells which were then subjected to DSS treatment or starvation, as in above studies. Effects on claudin-2 expression and cleaved caspase-3 was determined. As shown in Fig. 6c, d and Supplementary Fig. 5a–d, silencing of ATG16L expression significantly prevented degradation of claudin-2 protein from both, colitogen or starvation-induced stress and promoted cell death.

To further examine the causal role of claudin-2 in increased cell death in autophagy-inhibited IECs, we genetically inhibited claudin-2 expression using anti-human claudin-2 siRNA. Control and claudin-2 knockdown cells were subjected to DSS (2.5% w/v)

in complete culture medium and effects on cell viability was determined. As shown in Fig. 6e, claudin-2 KD significantly promoted the cell death in response to DSS treatment compared to control cells. We further confirmed this finding using HT29 cells stably expressing anti-claudin-2 (human) shRNA (HT29 cells). Claudin-2 knockdown (shRNA-HT29) cells showed significantly more cells death compared to control cells in response to the collagen-treatment (Fig. 6e–g). We also validated above findings using ex-vivo 3D-culture of the intestinal crypts from claudin-2 KO mice and WT littermates. Immunoblot analysis using the lysates from these crypt cultures, exposed to DSS (1% w/v for 12-h), showed a significant increase in the expression of cleaved caspase-3 expression in Claudin-2 KO mice compared to WT mice (Fig. 6h, i). We found a contrasting decrease in the expression of cleaved caspase-3 expression in the DSS-treated crypt culture from Villin-claudin-2 transgenic mice versus WT mice (Fig. 6h, i). Based on these findings, we postulated a sharp increase in cell death in IECs expressing the K218A mutant claudin-2 protein when subjected to stress. Immunoblot analysis was done for cleaved caspase-3 expression using the lysates from control and DSS-treated Caco-2 cells overexpressing full-length, K216A or K218A mutants. Indeed, we found significant increases in cleaved caspase-3 expression in cells overexpressing K218A mutant protein compared to the cells expressing full-length or K216A mutant protein, in response to the DSS treatment (Fig. 6j). Taken together, above findings informed us for a novel role of claudin-2 in promoting survival in IECs and strongly supported integration between autophagy and intestinal claudin-2 expression under the conditions of stress/ injury.

**In IBD patients, claudin-2 expression is dysregulated and colocalize with P62/SQSTM1 and LC-3.** Above data suggested that claudin-2 degradation helps protect the IECs from cell death via selective autophagy. However, claudin-2 expression is upregulated in IBD[12]. Of note, impaired autophagy is a hallmark of IBD[9,37]. Hence, we further examined whether upregulated claudin-2 in IBD could be a response to impaired autophagy. In this regard, we co-immunostained biopsies from the IBD patients with claudin-2 and P62/SQSTM1 or claudin-2 and LC-3. As shown in Fig. 7a, b, P62/SQSTM1 and LC-3 expressions were sharply upregulated in the biopsies from IBD patients compared to the normal colon. We further noted that a specific fraction of claudin-2 protein in IBD patient biopsies was co-localized with P62/SQSTM1 and LC-3 compared to the normal colon. Taken together, these data suggested that impaired autophagy in IECs under conditions of IBD may hinder autophagy-mediated degradation of claudin-2, which in turn may promote cell death and thus epithelial injury (Fig. 7c).

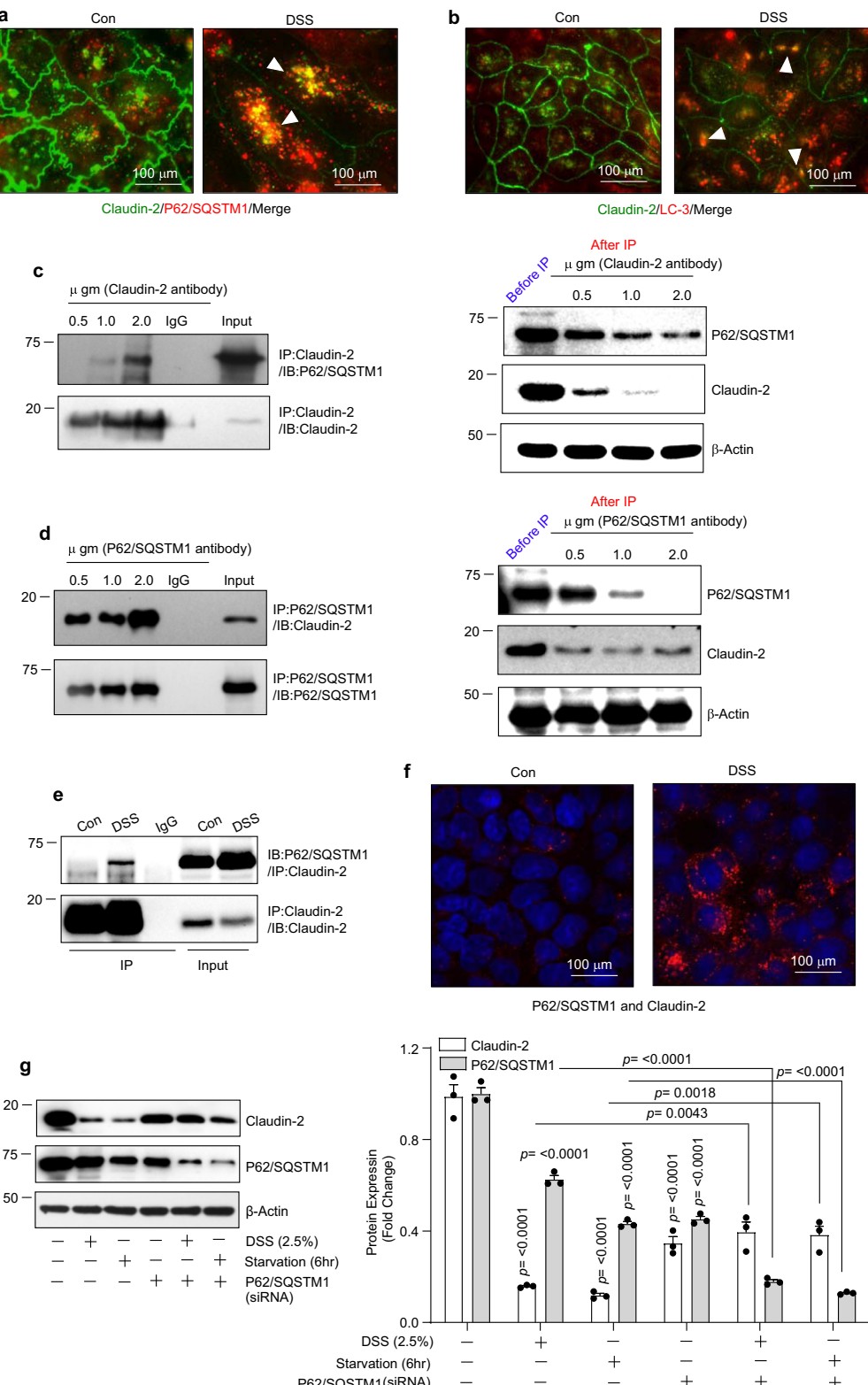

**Fig. 3 P62/SQSTM1 physically associates with claudin-2. a** Representative images of immunofluorescent colocalization of claudin-2 and P62/SQSTM1 in control and DSS-treated Caco-2 cells. Arrows indicate colocalization. **b** Immunofluorescent analysis of claudin-2 and LC-3 expression in control and DSS-treated cells. Arrows indicate colocalization. **c** Immunoprecipitation of claudin-2 using progressively increasing amount of anti-claudin-2 antibody. The immunoprecipitant and the flow-through samples were immunoblotted for claudin-2 and P62/SQSTM1 proteins. **d** Immunoblot analysis of claudin-2 and P62/SQSTM1 protein in immunoprecipitant derived using anti- P62/SQSTM1 antibody

for immunoprecipitation and using lysates from Caco-2 cell. **e** Immunoblot analysis for P62/SQSTM1 proteins in immunoprecipitants from control and DSS-treated Caco-2 cell lysates using anti-claudin-2 antibody. **f** Representative images of proximity-ligation analysis for claudin-2 and P62/SQSTM1 in Caco-2 cells subjected to the DSS treatment. **g** Immunoblotting and densitometric analysis of claudin-2 expression in Caco-2 cells where P62/SQSTM1 expression was genetically silenced and then cells were subjected to DSS treatment and/or nutrient starvation. $n = 3$ independent experiments. Results are presented as means ± SEM. *P* values from one-way ANOVA followed by Tukey's post hoc test.

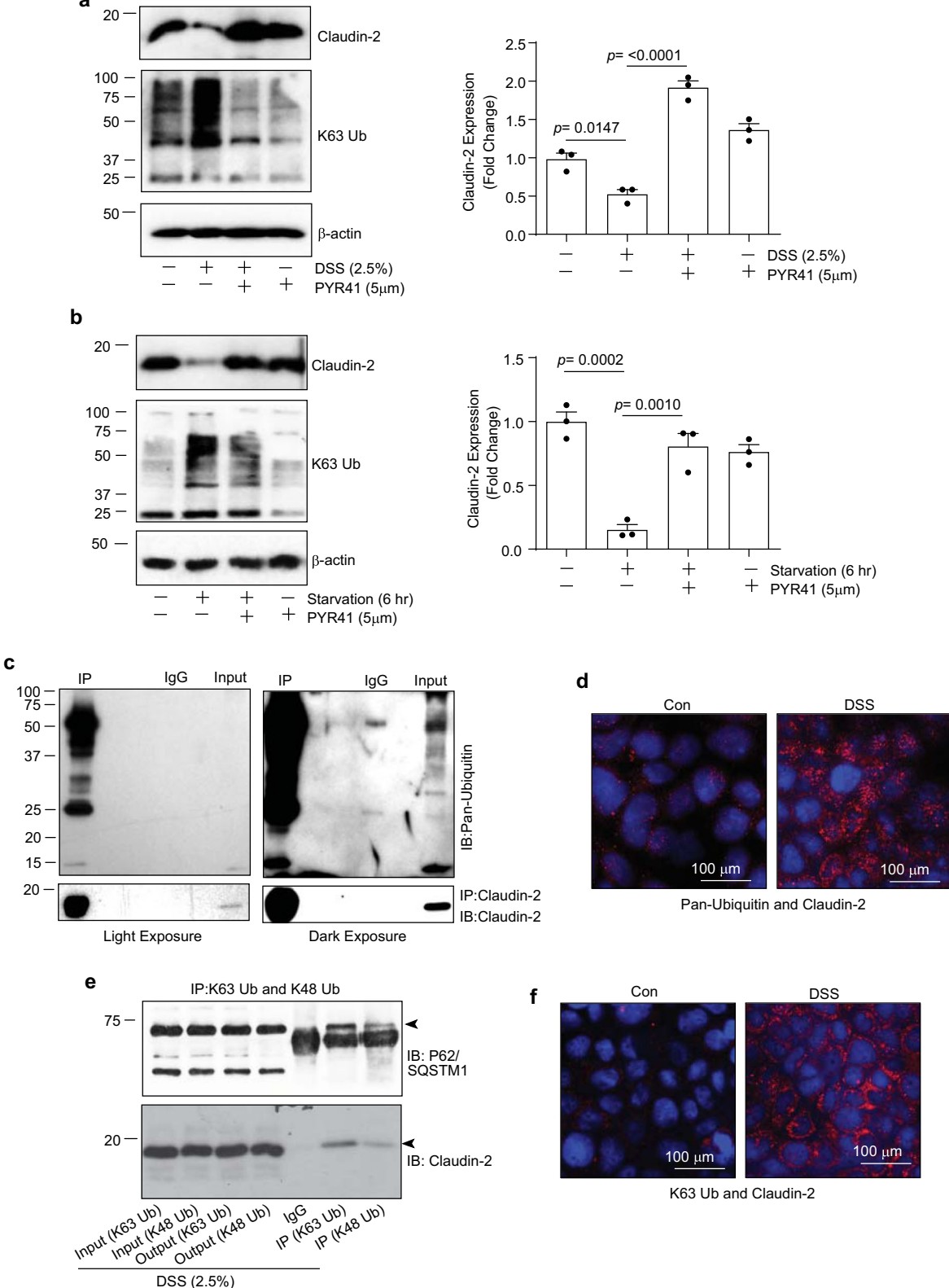

**Fig. 4 Colitogen or nutrient stress induce K63-linked ubiquitination of claudin-2. a**, **b** Immunoblot analysis of claudin-2 and K63-Ubiquitin using total cell lysate from Caco-2 cells subjected to DSS treatment or starvation with/without PYR41, a cell-permeable ubiquitin E1 ligase inhibitor (5 μM; $n = 3$ independent experiments. Results are presented as means ± SEM. *P* values from one-way ANOVA followed by Tukey's post hoc test). **c** Immunoprecipitants pulled down using anti-claudin-2 antibody and Caco-2 cells lysates were immunoblotted using the pan-ubiquitin or claudin-2 antibody. **d** Immunofluorescent imaging of proximity-ligation assay (PLA) using claudin-2 and Pan-ubiquitin antibody. **e** Immunoprecipitation was done using Caco-2 cell lysates and antibodies against K63-linked and K48-linked ubiquitin followed by immunoblotting using anti-claudin-2 or P62/SQSTM1 antibody. **f** Representative images of PLA analysis using antibodies against claudin-2 and K63-Linked ubiquitin in Caco-2 cells subjected to DSS treatment.

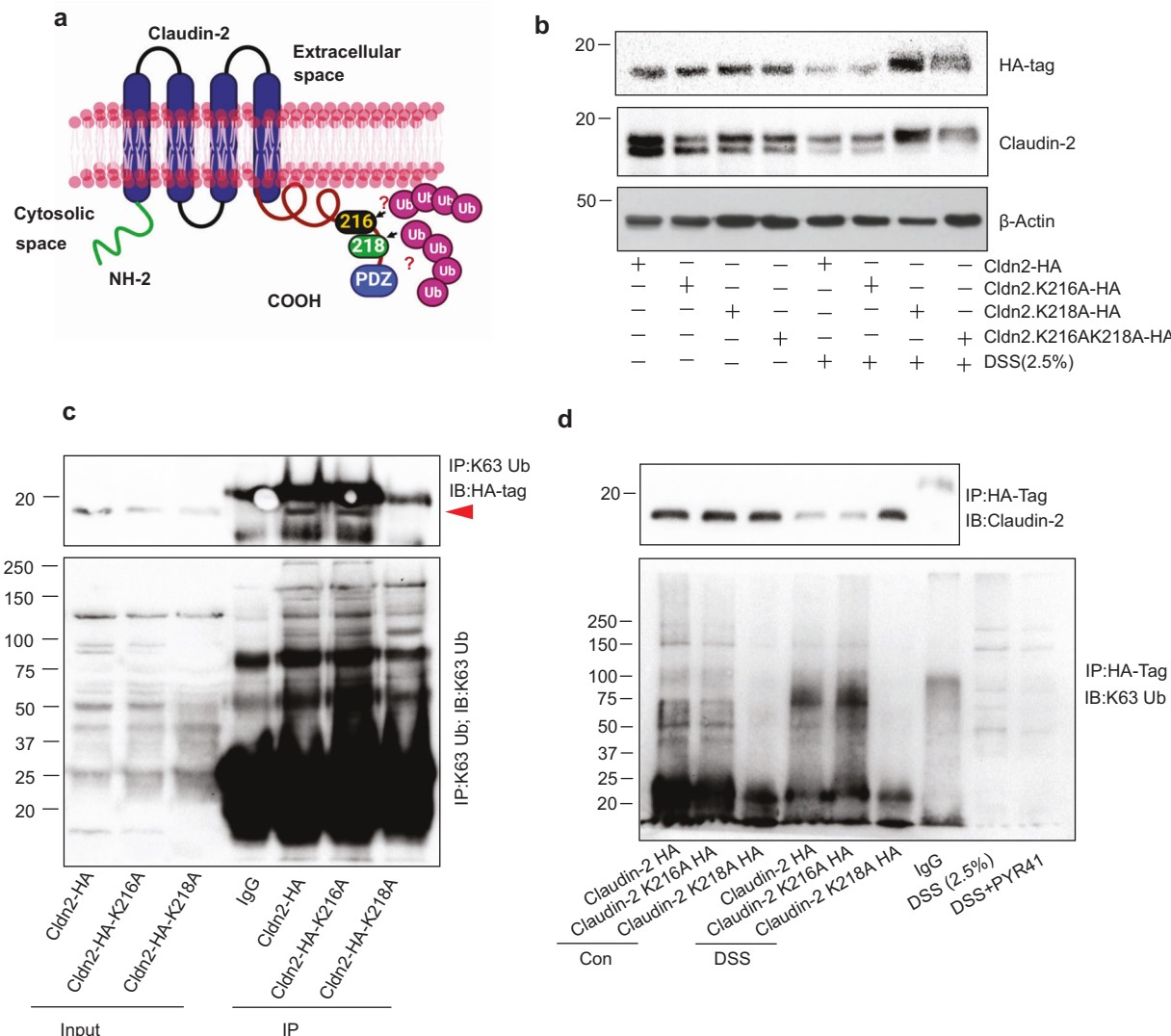

**Fig. 5 Lysine (K) 218 in claudin-2 cytoplasmic tail undergoes K63-linked ubiquitination in stressed IECs. a** Schematic representation of lysine positions in the cytoplasmic tail of claudin-2 protein which were substituted with alanine; **b** Immunoblot analysis using anti-claudin-2 and HA-tag antibodies, and lysates from Caco-2 cell transiently overexpressing Claudin-2-HA(Cldn2-HA), Claudin-2K216A (Cldn2K216A-HA), Claudin-2K218A-HA (cldn2K218A-HA) or both mutations Claudin-2K216AK218A-HA (Cldn2K216AK218A-HA) constructs. Effect of DSS treatment was determined compared to the untreated control cells. **c** Immunoprecipitation using anti-K63-linked ubiquitin using lysates from Caco-2 cells overexpressing full-length Cldn2-HA or claudin-2 mutant (Cldn2K216A or Cldn2K218A-HA) cDNA constructs. Immunoblotting was done with anti-K63-linked ubiquitin and anti-HA-tag antibody. **d** Immunoprecipitation using anti-HA-Tag antibody and lysates from Caco-2 cell overexpressing Cldn2-HA, Clan2K216A-HA or Cldn2K218A-HA constructs and treated with DSS. Immunoblotting was done using K63-linked ubiquitin antibodies, respectively.

## Discussion

An impaired mucosal barrier function is a hallmark of the onset and perpetuation of IBD[38,39]. Apart from genetic, environmental, and dietary factors, mucosal injury/repair regulates intestinal barrier integrity. Autophagy plays a key role in regulating intestinal epithelial homeostasis including mucosal healing and barrier integrity[40]. Accordingly, dysregulated autophagy associates with poor prognosis in both, mice, and humans in the context of IBD[10,37,41,42]. Our current findings that targeting of claudin-2 protein by selective autophagy promotes survival in stressed IECs provides a novel perspective for how autophagy regulates mucosal barrier integrity. Our findings also help explain previous findings that mice overexpressing claudin-2 in their intestinal epithelium are protected from DSS-colitis while claudin-2 KO mice develop severe colitis[17,18]. In this regard, DSS treatment causes injury to the intestinal epithelium to induce colitis which was supported by our data that DSS treatment induces DNA damage and apoptosis in

IECs or mouse colon. Of note, the DSS-colitis is often used as the model of inflammation-associated injury/repair[43]. We concede that the in vitro use of DSS treatment does not reflect the in vivo conditions of colitis as immune components are lacking. However, current study was undertaken to examine epithelial-intrinsic effects of claudin-2 expression under conditions of colitis/stress. Recent studies have used similar in vitro treatment of IECs by DSS to model colitis-induced epithelial injury[44–47]. However, both, DSS treatment and starvation resulted in a similar fate for claudin-2 and IECs survival, which suggest that the autophagy-mediated regulation of claudin-2 may not be limited to IBD. Taken together, above findings attest to a context-specific role of claudin-2 in regulating intestinal homeostasis and suggest that selective autophagy targets claudin-2 in IECs as a protective switch under acute adverse/pathological conditions.

Our data that inhibiting autophagy resulted in accumulation of claudin-2 protein even in control cells suggests that autophagy-

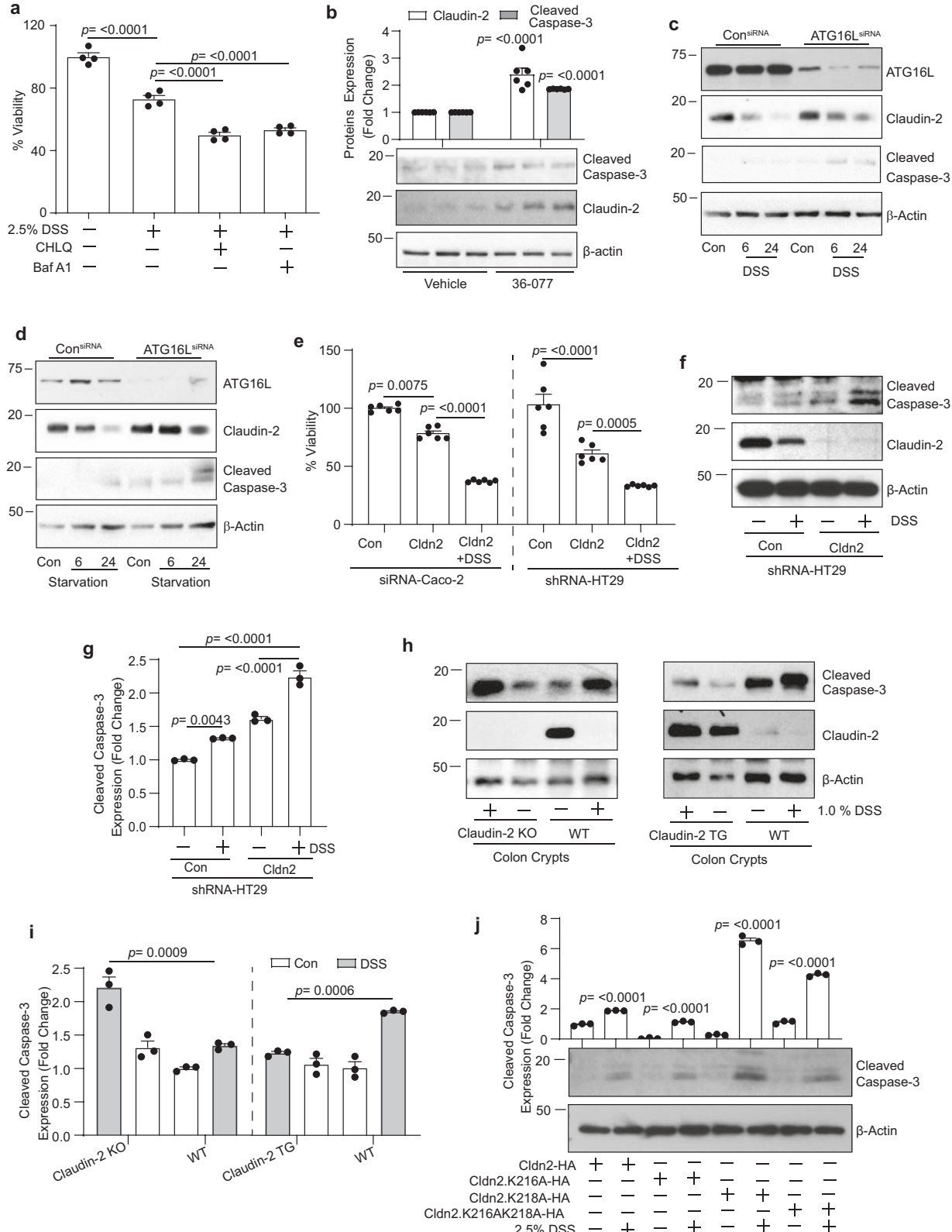

mediated claudin-2 targeting is a fundamental aspect of the autophagy-mediated regulation of the gut barrier functions. Our additional data that inhibiting endosomal or proteasomal trafficking does not lead to similar levels of accumulation of claudin-2 protein, either in control or treated cells further support the proposition that stress-induced autophagy flux induces claudin-2

protein degradation. Our data gains support from the study by Nighot et al. as it also demonstrated that starvation-induced autophagy degrades claudin-2 protein[20]. A follow-up study by this group demonstrated that AP2M1 mediates autophagy-induced claudin-2 degradation through endocytosis[48]. Clathrin and Rab14 also regulate vesicular trafficking of claudin-2[23,24,49]. However, the

**Fig. 6 Inhibiting autophagy inhibits claudin-2 protein degradation and promotes cell death in stressed IECs. a** Cell viability in Caco-2 cells subjected to DSS treatment with/without chloroquine and Bafilomycin. $n = 4$ independent experiments. Results are presented as means ± SEM. $P$ values from one-way ANOVA followed by Tukey's post hoc test. **b** Immunoblotting and densitometric analysis using total colon lysate from WT mice, untreated or treated with 36-077, a potent and selective inhibitor of autophagy. $n = 3$ mice/group and results are presented as means ± SEM. $P$ values from Student's $t$ test. **c, d** Immunoblotting for ATG16L, claudin-2, and cleaved caspase-3 using lysates from Caco-2 cells transiently transfected with anti-human ATG16L or control siRNA and treated with DSS (2.5%) and starvation. **e** Cell viability assay using transient transfected claudin-2 siRNA in Caco-2 or HT29 cells stably overexpressing anti-human claudin-2 shRNA, and control cells. $n = 6$ independent experiments. Results are presented as means ± SEM. $P$ values from one-way ANOVA followed by Tukey's post hoc test. **f, g** Immunoblot analysis for cleaved caspase-3 and claudin-2, and densitometric quantitation. $n = 3$ independent experiments. Results are presented as means ± SEM. $P$ values from one-way ANOVA followed by Tukey's post hoc test. **h, i** Immunoblot analysis using 3D-cultured freshly isolated mouse colon crypts from WT, Villin-claudin-2 transgenic and claudin-2 knockout mice subjected to DSS treatment (1% DSS w/v), $n = 3$ independent experiments. Results are presented as means ± SEM. $P$ values from one-way ANOVA followed by Tukey's post hoc test. **j** Representative immunoblots analysis of cleaved caspase-3 expression and densitometry using lysates from Caco-2 cells transiently transfected with full-length or claudin-2 mutant constructs and subjected to DSS treatment. $n = 3$ independent experiments. Results are presented as means ± SEM. $P$ values from one-way ANOVA followed by Tukey's post hoc test.

precise mechanism how membrane-localized claudin-2 is sequestered for its autophagy-mediated degradation under stress conditions remain unclear and part of our ongoing studies.

The study by Nighot et al. also concluded that the autophagy-mediated claudin-2 degradation is an adaptive response as it promotes barrier integrity in IECs[20]. We have found a similar decrease in claudin-2 expression in the IECs or colonic crypts when subjected to nutrient starvation. We however also found that starvation increases the expression of cleaved caspase-3, suggesting increased apoptosis. These data are in accordance with other reports that starvation induces cell death[50]. These data along with published data that claudin-2 expression protects mice from DSS-colitis led us to investigate whether autophagy-mediated claudin-2 degradation may help promote barrier integrity in the IECs from two different perspectives: (1) Claudin-2 is also named as the leaky claudin due to its role in paracellular cation transport[16,51]. Thus, a decrease in the membrane-integral claudin-2 protein expression would result in decreased paracellular permeability and hence increased barrier integrity. This postulation is supported by the published data in ref. [20]; and (2) Autophagy-mediated claudin-2 degradation in the stressed IECs would promote cell survival, which in turn would promote barrier integrity. In this regard, our data using both in vivo and in vitro studies, and using different colitogen showed down-regulation of claudin-2 protein expression, which was preventable by inhibiting autophagy. Our data that DSS-colitis induced sharp decreases in claudin-2 expression contrasts with some published studies that have shown an increase in claudin-2 expression in mice subjected to DSS-colitis[52,53]. In our opinion, such a discrepancy could be the result of the use of proximal versus distal colon segments for the immunoblotting studies. In our studies, we have used only distal colon known to be majorly impacted during DSS-colitis. Moreover, in our studies using IECs and crypt culture, we consistently observed a sharp decrease in claudin-2 expression. Overall, we believe that the outcome from the current study is unique, which suggests a context-specific role of claudin-2 in regulating IEC homeostasis and barrier integrity.

Similar to inhibiting autophagy, inhibiting ubiquitination also prevented starvation or colitogen-induced claudin-2 degradation while promoting cell death. Ubiquitination marks the proteins for degradation, including autophagy-mediated targeting[54]. Accordingly, Co-IP studies using pan-ubiquitin or claudin-2 antibodies demonstrated that claudin-2 protein in our studies is poly-ubiquitinated. Mechanistically, for targeting of the proteins by selective autophagy, P62/SQSM1 uses the ubiquitin-associated domain to recognize the ubiquitinated substrates and the LC-3-interacting region (LIR) motif for association with the LIR docking site (LDS) on LC-3[54,55]. Using Co-IP, proximity-ligation assay, and antigen-depletion assays, we have demonstrated that claudin-2 binds with P62/SQSM1. In this relation, K63-linked

ubiquitination at the lysine site is crucial for P62/SQSM1-mediated selective autophagic protein targeting, while K48-linked ubiquitination has been linked with proteasome-mediated protein degradation[56]. Our analysis determined that claudin-2 associates primarily with the K63-linked ubiquitin under our experimental conditions. We do see a minor association of claudin-2 with K48-linked ubiquitin however its significance remains currently unclear. Importantly, mutating the K218 in claudin-2 cytoplasmic tail not only prevented stress-induced claudin-2 targeting but also inhibited the physical association of claudin-2 with P62/SQSM1 and K63-linked ubiquitin, further supporting the postulation that claudin-2 is targeted by selective autophagy in stressed IECs. Our data that silencing of P62/SQSM1 expression ameliorates stress-induced claudin-2 degradation further supported its causal role in autophagy-mediated claudin-2 degradation. Additional studies demonstrating that the UBA domain of P62/SQSTM1 is important for the autophagy-mediated degradation of claudin-2 in stressed IECs would have further strengthened the proposition that P62/SQSTM1 mediates autophagy-dependent claudin-2 degradation. However, these studies are beyond the scope of the current manuscript and are part of our ongoing studies. Taken together, above data supported the postulated role of autophagy-mediated claudin-2 targeting in promoting IECs survival.

Published studies, including from our laboratory, have shown an upregulated claudin-2 expression in IBD patients. However, molecular mechanisms underlying this increase in claudin-2 levels and its causal significance remain unclear especially considering the context-specific outcome from the murine models of IBD using mice genetically manipulated for claudin-2 expression[17–19]. Notably, claudin-2 has been implicated in regulating paracellular ion transport, proliferation, and migration[14–17]. Also, claudin-2 expression has been shown in the immune cells and fibroblasts[57,58]. Considering our current data and the fact that autophagy is impaired in IBD patients, we further examined whether upregulated claudin-2 levels in IBD patients is due primarily to inhibition of autophagy-targeted claudin-2 degradation. Immunofluorescence analysis using the IBD-patients biopsies indeed showed dysregulated claudin-2 localization in vesicles which co-localized with P62/SQSM1 and LC-3. Also, inhibiting autophagy in mice using a highly specific and potent autophagy inhibitor further resulted in accumulation of claudin-2 and the concomitant increase in the expression of cleaved caspse-3. Similarly, inhibiting ATG16L expression, mutated in IBD patients[8], in IECs prevented the colitogen/starvation-induced claudin-2 degradation and promoted cell death. Taken together, our data demonstrate a context-specific and causal integration between autophagy and claudin-2 in regulating intestinal homeostasis and inflammation, which may be of clinical interest in the management of inflammatory intestinal diseases.

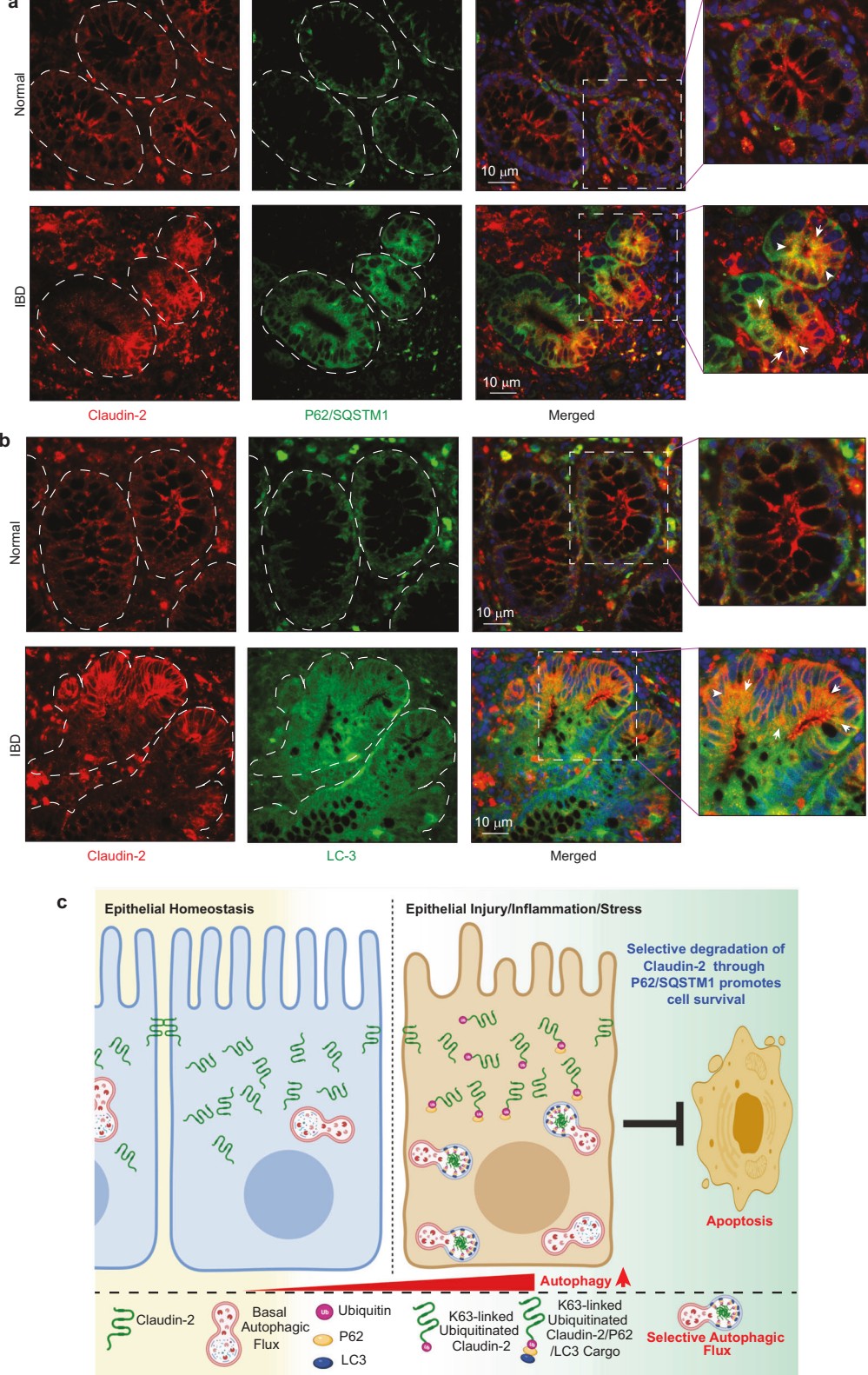

**Fig. 7 Claudin-2 expression in upregulated in IBD patient samples and co-localizes with autophagy markers. a** Immunofluorescence image of claudin-2 and P62/SQSMT1 using biopsies samples from IBD patients and insert represent the colocalization of claudin-2 and P62/SQSMT1. **b** Co-immunofluorescence analysis using of anti-claudin-2 and- LC-3 antibodies and insert represent the colocalization of claudin-2 and LC-3. **c** Graphical representation of the overall concept.

## Methods

**Materials and animals**. Dextran sodium sulfate (DSS) was purchased from TdB Labs (Sweden). The anti-claudin-2 antibodies were obtained from Invitrogen (Carlsbad, CA). The Lamp-1, cleaved PARP, Pan Ubiquitin, cleaved Caspase-3, and anti-H2AX antibodies were purchased from Cell Signaling (Boston, MA). The E-cadherin antibody was from BD Biosciences (Franklin Lakes, NJ). Antibodies for β-actin, P62/SQSM1 and LC-3 were obtained from Sigma-Aldrich (St. Louis, MO). Bafilomycin A1 and Chloroquine were also purchased from Sigma-Aldrich (St. Louis, MO). Supplementary Table 3 shows the list of antibodies and reagents used. Caco-2 and HT29 cells used in the current study were either purchased from ATCC or acquired from collaborative laboratories and have been described previously[59]. Only low-cell passages and mycoplasma-free cells were used, and the identity of cell lines used for detailed studies was routinely verified by DNA fingerprinting. These cells were cultured in DMEM-high glucose containing 10% FBS, 100 U/ml penicillin, and 100 μg/ml streptomycin. All cell culture reagents were from Invitrogen (San Francisco, CA, USA). C57BL/6 mice were originally purchased from The Jackson Laboratory (Bar Harbor, ME) and bred in our facility under specific pathogen-free conditions. All mice studies were approved by the Institutional Animal Care and Use Committee (protocol no: 17-126-11FC), University of Nebraska Medical Center, Omaha, USA, and in all animal studies we adhered to the animal experimentation ethical guidelines.

**IBD patient samples**. All work was carried out under the Institutional Review Board approved guidelines, University of Nebraska Medical center Omaha, NE. All IBD patient samples were de-identified. Diagnosis of IBD was based on clinical symptoms and endoscopic findings and confirmed by a pathologist using histological examination.

**In vitro modeling of colitogen-induced intestinal epithelial cell injury**. Cells were seeded in Dulbecco's modified Eagle medium with high glucose (DMEM-High Glucose) medium supplemented with 10% FBS, 4 mM glutamine, 110 mg/L sodium pyruvate, and 1% antibiotic cocktail. Confluent cells (75–90%) or mouse crypt culture were treated with colitogen (2.5% DSS or 2.5 mM TNBS or as described for a specific experiment) for different times, as noted for the specific study.

**Nutrient starvation**. Cells were grown for a minimum 60% confluence in a complete culture medium. Thereafter, cells were washed twice in Earle's balanced salt solution (EBSS) that was acclimatized in the culture incubator and cultured thereafter in EBSS for the desired period of starvation[20].

**In vivo induction of intestinal epithelial injury**. C57BL/6 mice (8–10 weeks old) were subjected to DSS-induced acute colitis using the published protocol. Briefly, we used 2.5% (w/v) DSS (MW = 36–50 kD, TdB Labs, Sweden) in drinking water for 7 days and then mice were sacrificed, and organs/tissues were collected[17].

**Cell death assay**. Cell death was determined by Cell Titer assay (MTT assay; Promega, Inc.) using the manufacturer's protocol. Briefly, 10,000 cells were plated in 96-well plates. Twenty-four hours after plating, cells were exposed to 2.5% DSS or starvation in a regular culture medium. The Cell Titer reagent was added 4 h before taking the final values.

**siRNA-mediated gene silencing**. Caco-2 cells were seeded in six-well plates ($0.2 \times 10^6$/well). Cells were transfected with anti-human ATG16L siRNA (s30070), claudin-2 (s225076), P62/SQSM1 siRNA (s16962), and control siRNA (4390843) using TurboFect Transfection Reagent (Thermo-scientific# R0533) as per manufacturer's protocol. After 24 h of transfection, cells were treated with 2.5% DSS or subjected to starvation for 6 and 24 h and then collected for immunoblot analysis.

**Immunoblotting, immunofluorescence, and immunohistochemistry**. To investigate the protein expression, immunoblot analyses were done. Protein samples were prepared using RIPA lysis buffer containing the protease and phosphatase inhibitor cocktails (Thermo Fisher Scientific). Whole-cell lysates (20 μg) were subjected to immunoblotting and then transferred to the PVDF membrane (Bio-Rad). Membranes were incubated with respective primary antibody overnight at 4 °C, then probed with HRP-conjugated secondary antibody. Signals was visualized using Clarity ECL (enhanced chemiluminescence). Signal intensities were quantified using Image Lab 6.1 software (Bio-Rad).

To determine the subcellular location of claudin-2 expression and its colocalization with other proteins under investigation, immunofluorescent analysis was performed using 4% (v/v) paraformaldehyde-fixed cells. Cells were subjected to permeabilization with 0.5% (v/v) Triton X-100 and blocking in 2.5% (v/v) normal goat serum. Anti-claudin-2 (1:500), -Lamp-1(1:500), -LC-3 (1:800), and -P62/SQSM1 (1:1000) antibodies were used as primary antibodies, and FITC- and Cy3-conjugated anti-rabbit/mouse IgG antibodies (Jackson immune-research) were used as secondary antibodies containing 4',6-diamidino-2-phenylindole (DAPI, Invitrogen) for nuclear staining. The paraffin-fixed slides of human and mouse colon sections were dewaxed and hydrated, and antigen unmasking was done using Tris-EDTA buffer (10 mM Tris Base and 1 mM EDTA at pH 9) in a delocking chamber. Sections were quenched with 3% $H_2O_2$ solution and blocked with 5% normal goat or horse serum. Tissue sections were incubated with specific primary antibodies overnight at 4°C in a humidified chamber. FITC- and Cy3-conjugated anti-rabbit/mouse IgG antibodies (Jackson immunoResearch) were used as secondary antibodies for immunofluorescence analysis. The ABC and DAB kits (Vector Lab) were used for immunohistochemistry to develop color after incubating respective primary antibodies overnight at 4°C. Image acquisition was done using Nikon's ECLIPSE Ti-S microscope processed by Adobe Photoshop CS (Adobe Systems Inc. San Jose, CA).

**Immunoprecipitation assay**. To examine the physical interaction of claudin-2 with other proteins under investigation, total cell lysates were prepared in IP lysis buffer (50 mM Tris-HCl, pH 7.5, 150 mM NaCl, 0.5% NP-40) and quantified. In total, 1 mg of total cell lysates was subjected to immunoprecipitation using protein-G Dynabeads (Invitrogen) and appropriate antibody to detect the physical interaction. Immune complexes were detected using horseradish peroxidase-conjugated secondary antibodies and Clarity ECL.

**Proximity-ligation assay**. To examine claudin-2 interaction to other proteins that included P62/SQSM1, ubiquitin, and K63-linked ubiquitin, we performed a proximity-ligation assay. We used Duolink PLA Multicolor Reagent Pack (# DUO92101-1KT Sigma) and followed the manufacturer's guidelines. Different species of antibodies were paired for detecting protein-protein interaction by proximity-ligation assay.

**Site-directed mutagenesis**. To mutate the desired lysine (K216 and K218) in claudin-2 cytoplasmic tail to alanine, we used PCR amplification using the full-length claudin-2 cDNA as a template and custom primers. In brief, CLDN2-K216A, CLDN2-K218A and CLDN2-K216AK218A constructs were created using mutant reverse primers, CLDN2.RP.K216A, CLDN2.RP.K218A and CLDN2.RP.K216A.K218A respectively. All constructs were amplified using the same forward primer named CLD2.H.COM.FP primer. All constructs were amplified using a common reverse prime CLDN2.RP, which covered the remaining C-terminal sequence of claudin-2. To add a C-terminal HA-tag, all constructs were further amplified using a reverse CLDN2.HA.RP primer (All primer sequences are described in Supplementary Table 2). The amplified products were cloned in a pCDNA3 vector in HindIII and XhoI sites by homologs recombination, as per the manufacturer's instructions (In-Fusion Snap Assembly Master Mix; # 638947). All clones were verified for appropriate sequence by sequencing.

**Statistics and reproducibility**. At least three individual experiments were performed in all the studies unless stated otherwise. Mean value with the standard error were used for independent Student $t$ tests or one-way ANOVA, and corrections for multiple comparisons were made using Tukey's multiple comparison tests in Prism 9.5 (GraphPad Software, Inc.). A $P$ value <0.05 was defined as statistically significant.

**Reporting summary**. Further information on research design is available in the Nature Portfolio Reporting Summary linked to this article.

## Data availability

Data sharing is not applicable to this article as no datasets were generated or analyzed during the current study. Supplementary Figs. 1–5 and Supplementary Tables 1–3 have been provided in the supplemental information pdf file. Supplementary Fig. 6 in the supplemental information pdf file provides all uncropped western blot images with size markers. Numerical source data for Figs. 1, 2, 3, 4, and 6 have been provided in Supplementary Data 1. Numerical source data for Supplementary Figs. 1–5 have been provided in Supplementary Data 2.

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

## Acknowledgements

We thank Dr. Howard Fox, MD, PhD, Senior Associate Dean of Research & Development, Professor, Department of Neurological Sciences, and Director, Center for Integrative & Translational Neuroscience, University of Nebraska Medical Center for sharing the LC3-GFP-RFP encoding stable HEK293 cells. This work was supported in part by the funds from VA-merit award (BX002761) and National Institute of Health RO1 grant funding (DK124095; to A.B.S.), and VA-merit award (BX002086) and National Institute of Health RO1 grant funding (CA250383; to P.D.).

## Author contributions

Conceptualization: A.B.S., R.A., and B.K.; data production, analysis, and investigation: R.A., B.K., R.L.T., G.A.T., and A.B.S.; writing, review, and editing: A.B.S., P.D., and R.A.; supervision: A.B.S.; funding acquisition: A.B.S. and P.D. All authors have read and agreed to the published version of the manuscript.

## Competing interests

The authors declare no competing interests.
