## [Peer Review File · Communications Biology]

Reviewers' comments:

Reviewer #1 (Remarks to the Author):

In this paper Ahmad et al. showed that the treatment of intestinal epithelial cells with colitoge or upon starvation decreases the plasma membrane protein Claudin-2, and it relies on autophagy and selective autophagy receptor p62. They claimed that Lys218 of Claudin-2 is ubiquitinated at Lys63-linkage upon the treatment of colitoge, which is indispensable for downregulation of Claudin-2. The defect of this pathway caused activation of caspase both in vitro and in vivo. Although some of the results are interesting, there are several scattered points where the interpretation of the experimental results is not well founded. Furthermore, the experiments are insufficient to prove the authors' claims.

1. Claudin-2 is a plasma membrane protein and to undergo degradation by ATG16L or p62-dependent autophagy (macroautophagy), Claudin-2 should be pulled from the plasma membrane or vesicles containing Claudin-2 should be formed from the plasma membrane. There is no experimental approach to this point, and it is not mentioned even in the Discussion section.
2. To prove p62-dependent degradation of Claudin-2, knockdown experiments targeting p62 and immunoprecipitation assays with a p62 mutant deleting the UBA domain are required.
3. The results with PR-619 were incomprehensible. The PR-619 treatment would cause an increase in ubiquitinated proteins since it is an inhibitor of broad range of deubiquitinating enzyme inhibitor. Nevertheless, the authors appear to be using PR-619 as an inhibitor of ubiquitination. Such experiments would normally be suitable for the use of a cell membrane-permeable ubiquitin activating enzyme (E1) inhibitor (e.g., UBEI-41). If the authors use deubiquitination or ubiquitination inhibitors, immunoblots with ubiquitin antibody should be presented to ensure their effectiveness.
4. The data shown in Fig. 4E-F merely show that there is Claudin-2ubiquitylated at the K63-linkage, but do not rule out ubiquitination at other linkages such as K48.
5. Fig. 5C-E is also difficult to interpret: if immunoprecipitation is performed with a K63-linkage-specific ubiquitin antibody and then the immunoprecipitates are analyzed by immunoblot with Claudine-2 antibody, the Claudine-2 detected should be ubiquitinated one. However, the detected one appears to be free Claudin-2, which is not ubiquitinated.
6. Fig. 2G and H does not mean autophagy-flux by mere blotting of LC3; autophagic flux should be verified using inhibitors such as Baf. A1.
7. Fig. 2A, why does actinomycin D treatment cancel the effect of DSS (Claudin-2 decrease)?

Reviewer #2 (Remarks to the Author):

This manuscript mainly studied the role of Claudin-2 in the DSS-induced autophagy, and found that DSS/starvation induces the degradation of Claudin-2 by P62/LC3-dependent manner. Authors also identified the specific ubiquitination site of Claudin-2. Overall, it presents a novel target of P62-dependent selective autophagy, and demonstrates that the degradation of Claudin-2 is associated with cell death induced by DSS/starvation.

Concerns:

- 1) The association between degradation of Claudin-2 and cell death is missing. What is the mechanism of cell death induced by degradation of Claudin-2? Authors need to further explore the mechanism of Claudin-2 degradation and cell death.
- 2) The quality of data is poor. For example, in Fig 4F, there isn't obvious increasment of ubiquitination of Claudin-2 after DSS treatment; In Fig 5D, there isn't ubiquitination of Claudin-2, and arrow indicated band might be none-specific band; In Fig 5E, there isn't any difference in ubiquitination of Claudin-2 between wild-type and mutant Claudin-2. Authors need to present data with higher quality, and some of the present data aren't convincing.
- 3) What's the role of Claudin-2 in DSS-induced cell death? promotion or prevention? As in Fig 6G, authors showed that over-expression of Claudin-2 prevented cell death. While in Fig 6H, K218 mutant of Claudin-2 (stable mutant) promote cell death. So the results are confusing.
- 4) Since authors have Claudin-2 transgenic mice and KO mice, it would be important to examine the role of Claudin-2 in cell death by in vivo model, for example, DSS-induced colitis model.

Reviewer #3 (Remarks to the Author):

The authors investigated the interaction between Claudin2 and P62/SQSTM1 and found that DSS can affect the ubiquitination degradation of Claudin2 through autophagy, which has a certain research value, but there are still some minor problems :

- 1.It is still controversial to directly treat cells with DSS to simulate chemically induced colitis model, and the mechanism of DSS-induced colitis animal model has not been fully clarified at present. It is suggested that the authors alternatively choose IECs treated with complex inflammatory factors (TNF- α and IFN- γ , etc.) for further verification.
- 2.Several WB bands of β -Actin show multiple bands (as shown in Fig.1A and 1B). Please provide the original pictures and explanations.
- 3.In Fig.3C and 3D, it is suggested to add the Input group in co-IP.
- 4.As suggested by the authors, the elevated level of claudin2 in IBD patients in previous studies is inconsistent with the DSS-induced mice and IEC in this article, and it is recommended to add human IBD samples to verify the levels of claudin-2, autophagy, and related molecules.

We are submitting the revised version of our original article (COMMSBIO-22-3066). Due to the extensive revisions and COVID-related supply chain issues and personal sickness, it took longer than anticipated in submitting this revised article. We would like to sincerely thank the reviewers for their time and effort in reviewing our manuscript and providing highly constructive comments and suggestions. We have modified the manuscript to the best of our capacity to address the reviewer's comments, which we believe has strengthened the manuscript. We hope that the revised article is considered suitable for publication in Communication Biology.

Below, please find the reviewer's comments summarized in black font, followed by our responses in blue font. Corresponding changes made to the manuscript text are shown in red font.

Reviewer #1 (Remarks to the Author):

Query#1. Claudin-2 is a plasma membrane protein and to undergo degradation by ATG16L or p62-dependent autophagy (macroautophagy), Claudin-2 should be pulled from the plasma membrane or vesicles containing Claudin-2 should be formed from the plasma membrane. There is no experimental approach to this point, and it is not mentioned even in the Discussion section.

Response: Reviewer has raised an excellent point. Published studies have demonstrated that claudin-2 protein from the membrane can be trafficked by Rab14, clathrin, and caveolin-1, which can be recycled back to the membrane or degraded in the lysosomes (PMID:26163137; 21660968; 22396724; 25694446; 30899070). Moreover, recently Ganapathy et al. showed that AP2M1 mediates autophagy-induced claudin-2 degradation through endocytosis (PMID: 34964704). We have discussed these possibilities in the discussion part in the revised manuscript.

Query#2. To prove p62-dependent degradation of Claudin-2, knockdown experiments targeting p62 and immunoprecipitation assays with a p62 mutant deleting the UBA domain are required.

Response: As per reviewer's suggestion, we now provide new data that genetic silencing of P62 expression (siRNA mediated) prevents both, DSS- and starvation-mediated downregulation of claudin-2 expression. These data are included in the revised manuscript (Figure 3G).

Query#3. The results with PR-619 were incomprehensible. The PR-619 treatment would cause an increase in ubiquitinated proteins since it is an inhibitor of broad range of deubiquitinating enzyme inhibitor. Nevertheless, the authors appear to be using PR-619 as an inhibitor of ubiquitination. Such experiments would normally be suitable for the use of a cell membrane-permeable ubiquitin activating enzyme (E1) inhibitor (e.g., UBEI-41). If the authors use deubiquitination or ubiquitination inhibitors, immunoblots with ubiquitin antibody should be presented to ensure their effectiveness.

Response: We appreciate the reviewer's constructive criticism and as suggested by the reviewer, we repeated our studies using PYR41, a cell-permeable ubiquitin-activating enzyme inhibitor. We also performed analysis of K63-linked ubiquitination under the same setting. In the revised manuscript, we have provided the new data as Figure 4A and 4B.

Query#4. The data shown in Fig. 4E-F merely show that there is Claudin-2 ubiquitinated at the K63-linkage, but do not rule out ubiquitination at other linkages such as K48.

Response: As per reviewer's suggestion, we performed parallel co-immunoprecipitation using lysates from control and DSS-treated Caco-2 cells and antibodies specific for K48 and K63 linked ubiquitin. The outcome demonstrated robust association of claudin-2 with K63-linked, however also a minor association of claudin-2 with K-48 linked ubiquitin was also observed. We have added these data as Figure 4E and Supplementary Figure 3 in the revised manuscript.

Query#5. Fig. 5C-E is also difficult to interpret: if immunoprecipitation is performed with a K63-linkage-specific ubiquitin antibody and then the immunoprecipitates are analyzed by immunoblot with Claudine-2 antibody, the Claudine-2 detected should be ubiquitinated one. However, the detected one appears to be free Claudin-2, which is not ubiquitinated.

Response: We repeated these studies using the same claudin-2 mutant constructs. Caco-2 cells where these constructs were overexpressed were then subjected to the DSS-treatment. The overexpressed claudin-2 protein was immunoprecipitated using anti-HA-Tag antibody followed by immunoblotting with anti-claudin-2 and -K63 ubiquitin antibodies. The mutant claudin2K218A HA exhibited markedly reduced K63-linked ubiquitination of claudin-2. This data is provided in revised manuscript as Figure 5D.

Query#6. Fig. 2G and H does not mean autophagy-flux by mere blotting of LC3; autophagic flux should be verified using inhibitors such as Baf. A1.

Response: As per reviewer's suggestion, we analyzed autophagy flux using Baf A1 in Caco-2 cells subjected to the DSS treatment. The outcome has been added to the revised manuscript as Figure 2I)

Query#7. Fig. 2A, why does actinomycin D treatment cancel the effect of DSS (Claudin-2 decrease)?

Response: We appreciate this insightful comment. In our detailed analysis, we found that the DSS-treatment also affects claudin-2 mRNA expression. However, this effect is significantly lower compared to the effects of stress/colitogens on claudin-2 protein expression (Supplementary Figure 2A). Thus, the slight rescue of the DSS-mediated claudin-2 downregulation by Actinomycin-D treatment was in the line of our additional data.

Reviewer #2:

Query#1. The association between degradation of Claudin-2 and cell death is missing. What is the mechanism of cell death induced by degradation of Claudin-2? Authors need to further explore the mechanism of Claudin-2 degradation and cell death.

Response: Increased autophagy flux supports cell survival under stress conditions. Our data suggest that claudin-2 serves as a substrate for selective autophagy. Our data that cells overexpressing claudin-2 mutant unable to undergo autophagic degradation also undergo apoptosis supports that autophagic-degradation of claudin-2 promotes cells survival. Mechanistic details of whether claudin-2 degradation also promotes other cell survival mechanisms are plausible and part of our ongoing studies.

2) The quality of data is poor. For example, in Fig 4F, there isn't obvious increasment of ubiquitination of Claudin-2 after DSS treatment; In Fig 5D, there isn't ubiquitination of Claudin-2, and arrow indicated band might be none-specific band; In Fig 5E, there isn't any difference in ubiquitination of Claudin-2 between wild-type and mutant Claudin-2. Authors need to present data with higher quality, and some of the present data aren't convincing.

Response: We regret that the reviewer felt the data quality is poor and difficult to interpret. We have repeated these studies to obtain better quality and informative data and in revised manuscript we have replaced the previous figures with new data (Figures 4E, Figure 5D, and Supplementary Figure 3).

3) What's the role of Claudin-2 in DSS-induced cell death? promotion or prevention? As in Fig 6G, authors showed that over-expression of Claudin-2 prevented cell death. While in Fig 6H, K218 mutant of Claudin-2 (stable mutant) promote cell death. So the results are confusing.

Response: We appreciate reviewer's insightful comment and regret that it was not apparent from the discussion. Claudin-2 overexpression protects mice from DSS-induced colitis and epithelial cell death. As noted above, our working hypothesis is that under stress, claudin-2 is selectively degraded via autophagy to protect cells against stress-induced cell death. Thus, overexpression of claudin-2 would

promote the autophagic flux and associated cell survival. Thus, when claudin-2 is mutated to not participate in autophagy, cells will undergo apoptosis as overexposed mutant claudin-2 is not available for the autophagic degradation and thus promoting autophagy flux.

4) Since authors have Claudin-2 transgenic mice and KO mice, it would be important to examine the role of Claudin-2 in cell death by *in vivo* model, for example, DSS-induced colitis model.

Response: We have previously reported that claudin-2 transgenic mice are protected against DSS-colitis and associated epithelial cell death (PMID#24670427). Another group has reported that claudin-2 KO mice develop severe colitis and epithelial cell death when subjected to DSS-colitis (PMID# 23306855).

Reviewer #3:

Query#1. It is still controversial to directly treat cells with DSS to simulate chemically induced colitis model, and the mechanism of DSS-induced colitis animal model has not been fully clarified at present. It is suggested that the authors alternatively choose IECs treated with complex inflammatory factors (TNF- α and IFN- γ , etc.) for further verification.

Response: Several recent studies on intestinal inflammation have reported DSS-treatment as the tool to induce intestinal epithelial cell injury (some of them-PMID#: 35495925; 30476915; 17089061, and 29396428). By directly exposing epithelial cells to DSS *in-vitro* conditions, we create a similar milieu as in the *in-vivo* model. Moreover, we have also treated cells with TNBS, yet another colitogen. Treating IECs with complex inflammatory factors such as TNF- α and IFN- γ would not serve our purpose as it would involve inflammatory signaling. Apart, throughout the study, we have used starvation as complementary model of stress induced epithelial injury.

Query#2. Several WB bands of β -Actin show multiple bands (as shown in Fig.1A and 1B). Please provide the original pictures and explanations.

Response: We regret the double band of β -Actin pointed out by the reviewer. However, sometimes multiple bands do appear during immunoblot development using some β -Actin antibodies. We have provided improved blots to replace these blots (Figure 1A and 1B).

Query#3. In Fig.3C and 3D, it is suggested to add the Input group in co-IP.

Response: In these figures, the inherent goal is to analyze physical interaction by the immune-depletion method. For this, we used increasing claudin-2 or P62 antibody concentration to deplete the interacting partner in the lysate. We ran two different gels; one gel confirmed that immune depletion worked, and the second showed a similar decrease in the amount of both P62 and claudin-2 when we depleted either one. However, as per the reviewer's suggestions, we repeated the experiment and replaced Figures 3C and 3D in the revised manuscript.

Query#4. As suggested by the authors, the elevated level of claudin2 in IBD patients in previous studies is inconsistent with the DSS-induced mice and IEC in this article, and it is recommended to add human IBD samples to verify the levels of claudin-2, autophagy, and related molecules.

Response: Autophagy is dysregulated in IBD patients (PMID#17921695; 17435756), which may explain why claudin-2 is upregulated in IBD. We performed co-immunofluorescence using anti-claudin-2, LC3, and P62 antibodies in IBD patient samples. The outcome showed claudin-2 immunolocalization that is dysregulated from its membrane-tethered localization which co-localized with LC3 and P62. These results have been included in the revised manuscript (Figure 7A and 7B).

Reviewers' comments:

Reviewer #1 (Remarks to the Author):

The authors addressed most issues raised in first round review process except verification of whether a ubiquitin-associate domain of p62 is required for the autophagic degradation of claudin2. Analysis using mutants of p62 (e.g., Δ UBA domain) would strengthen the authors' model.

Reviewer #2 (Remarks to the Author):

The authors have addressed all my concerns.

Reviewer #4 (Remarks to the Author):

Accept with revision as proposed by author and comment on DSS method in discussion

We would like to thank the reviewers for their appreciation of the changes we made in revising our manuscript (COMMSBIO-22-3066A). All the reviewers had largely positive comments however some concerns lingered on. We have modified the manuscript again to address the remaining concerns and hope that the manuscript is now considered acceptable for publication.

Below, we have provided the detailed responses to the reviewer's comments in blue font. Corresponding changes in the manuscript are also in blue font.

Reviewer #1:

Critique: The authors addressed most issues raised in first round review process except verification of whether a ubiquitin-associate domain of p62 is required for the autophagic degradation of claudin2. Analysis using mutants of p62 (e.g., Δ UBA domain) would strengthen the authors' model.

Response: We thank the reviewer for the positive comments. We agree that additional analysis using mutants of P62/SQSTM1 (e.g., Δ UBA domain) could strengthen the manuscript. However, in our search for appropriate constructs for P62/SQSTM1 to perform the right experiment, we realized that the right constructs and tools are not available commercially. This would require construction of a full-length and Δ UBA domain depleted/mutated P62/SQSTM1 construct in our lab which may be time-taking. We plan to perform these analysis in our ongoing studies however feel that it is beyond the scope of this manuscript and does not add further to the conclusion that P62/SQSTM1 plays an essential role in colitis-induced degradation of claudin-2 protein.

Reviewer #2:

Critique: The authors have addressed all my concerns.

Response: We would like to thank the reviewer for his/her time and effort in reviewing the manuscript.

Reviewer #4:

Critique: Accept with revision as proposed by author and comment on DSS method in discussion

Response: We thank the reviewer for his/her time and effort in reviewing our manuscript. As per reviewer's suggestion we have added the limitation on the use of the DSS for *in vitro* studies of colitis in the discussion.